# PRETRAINING THE VISION TRANSFORMER USING SELF-SUPERVISED METHODS FOR VISION BASED DEEP REINFORCEMENT LEARNING

## ABSTRACT

The Vision Transformer architecture has shown to be competitive in the computer vision (CV) space where it has dethroned convolution-based networks in several benchmarks. Nevertheless, Convolutional Neural Networks (CNN) remain the preferential architecture for the representation module in Reinforcement Learning. In this work, we study pretraining a Vision Transformer using several state-of-the-art self-supervised methods and assess data-efficiency gains from this training framework. We propose a new self-supervised learning method called TOV-VICReg that extends VICReg to better capture temporal relations between observations by adding a temporal order verification task. Furthermore, we evaluate the resultant encoders with Atari games in a sample-efficiency regime. Our results show that the vision transformer, when pretrained with TOV-VICReg, outperforms the other self-supervised methods but still struggles to overcome a CNN. Nevertheless, we were able to outperform a CNN in two of the ten games where we perform a 100k steps evaluation. Ultimately, we believe that such approaches in Deep Reinforcement Learning (DRL) might be the key to achieving new levels of performance as seen in natural language processing and computer vision.

## 1   INTRODUCTION

Despite the successes of deep reinforcement learning agents in the last decade, these still require a large amount of data or interactions to learn good policies. This data inefficiency makes current methods difficult to apply to environments where interactions are more expensive or data is scarce, which is the case in many real-world applications. In environments where the agent doesn't have full access to the current state (partially observable environments), this problem becomes even more prominent, since the agent not only needs to learn the state-to-action mapping but also a state representation function that tries to be informative about a state given an observation. In contrast, humans, when learning a new task, already have a well-developed visual system and a good model of the world which are components that allow us to easily learn new tasks. Previous works have tried to tackle the sample inefficiency problem by using auxiliary learning tasks (Schwarzer et al., 2021b; Stooke et al., 2021; Guo et al., 2020), that try to help the network's encoder to learn good representations of the observations given by the environments. These tasks can be supervised or unsupervised and can happen during a pretraining phase or a reinforcement learning (RL) phase in a joint-learning or decoupled-learning scheme.

In recent years, self-supervised learning has shown to be very useful in computer vision, the increasing interest in this area has resulted in the appearance of new and improved methods that train a network to learn important features from the data using only the data itself as supervision. A common approach to evaluating such methods is to train a network composed of the pretrained encoder, with the parameters frozen, paired with a linear layer in popular datasets, like ImageNet. These evaluations have shown that these methods can achieve high scores in different benchmarks, which shows how well the current state-of-the-art methods are able to encode useful information from the given images without being task-specific. Additionally, it has been shown that pretraining a network using self-supervised learning (or unsupervised) adds robustness to the network and gives better generalization capabilities (Erhan et al., 2010).

Also recently, a new architecture for vision-based tasks called the Vision Transformer (ViT) (Dosovitskiy et al., 2020) has shown impressive results in several benchmarks without using any convolutions. This architecture presents much weaker inductive biases when compared to a CNN, which can result in lower data efficiency. But the Vision Transformer, unlike the CNNs, can capture relations between parts of an image (patches) that are far apart from each other, thus deriving global information that can help the model perform better in certain tasks. Furthermore, when the model is pretrained, using supervised or self-supervised learning, it manages to surpass the best convolution-based models in terms of task performance. Nonetheless, and despite these successes in computer vision these results are yet to be seen in reinforcement learning.

Motivated by the potential of the Vision Transformer, in particular when paired with a pretraining phase, and the increasing interest in self-supervised tasks for DRL, we study pretraining ViT using state-of-the-art (SOTA) self-supervised learning methods and use it as the representation module in a Deep RL algorithm. Consequently, we propose extending VICReg (Variance Invariance Covariance Regularization) (Bardes et al., 2022) with a temporal order verification task (Misra et al., 2016) to help the model better capture the temporal relations between consecutive observations. We named this approach Temporal Order Verification-VICReg or in short TOV-VICReg. While we could have adapted any of the other methods, we opted for VICReg due to its computational performance, simplicity, and good results in early experiments and metrics such as the ones presented in Section 7. After our empirical results in the Atari games, we present a small study of the pretrained encoders using several metrics to understand if they suffer from any representational collapse and also analyse the learned representations using similarity matrices and attention maps.

Our main contributions are:

- We propose a new self-supervised learning method which extends VICReg to capture the temporal relations between consecutive frames through a temporal order verification task, in Section 4.
- We pretrain a Vision Transformer using several SOTA self-supervised methods and our proposed method, and study them through metrics (Section 7), visualizations (Section 8) and fine-tuning in reinforcement learning ( Section 6), where we show that temporal relations learned by the model pretrained with our method contribute to a relevant increase in data efficiency.

## 2 RELATED WORK

**Pretraining representations**   Previous work, similarly to our approach, has explored pretraining representations using self-supervised methods which led to great data-efficiency improvements in the fine-tuning phase (Schwarzer et al., 2021b; Zhan et al., 2020) or superior results in evaluation tasks, like AtariARI (Anand et al., 2020). Others have pretrained representations using RL algorithms, like DQN, and transfer those learned representations to a new learning task (Wang et al., 2022).

**Temporal Relations**   Other works have explored learning representations that have temporal information encoded. ATC (Augmented Temporal Contrast) (Stooke et al., 2021) trains an encoder to compute temporally consistent representations using contrastive learning, and the ST-DIM (SpatioTemporal DeepInfoMax) (Anand et al., 2020) captures spatial-temporal information by maximizing the mutual information between features of two consecutive observations.

**Joint learning**   In recent years, adding an auxiliary loss to the RL loss, usually called joint learning, has become a common approach by many proposed methods. Curl (Srinivas et al., 2020) adds a contrastive loss using a siamese network with a momentum encoder. Another work studies different joint-learning frameworks using different self-supervised methods (Li et al., 2022). SPR (Schwarzer et al., 2021a) uses an auxiliary task that consists of training the encoder followed by an RNN to predict the encoder representation k steps into the future. PSEs (Agarwal et al., 2021a) combines a policy similarity metric (PSM), that measures the similarity of states in terms of the behaviour of the policy in those states, and a contrastive task for the embeddings (CME) that helps to learn more robust representations. PBL (Guo et al., 2020) learns representations through an interdependence between an encoder, that is trained to be informative about the history that led to that observation, and an RNN that is trained to predict the representations of future observations. Proto-RL (Yarats

et al., 2021) uses an auxiliary self-supervised objective to learn representations and prototypes (Caron et al., 2020), and uses the learned prototypes to compute intrinsic rewards which will push the agent to explore the environment.

**Augmentations**  While we only use augmentations in the pre-training phase, their use during reinforcement learning has also been studied. Methods like DrQ (Kostrikov et al., 2021) and RAD (Laskin et al., 2020) pair an RL algorithm, like SAC, with image augmentations to improve data efficiency and generalization of the algorithms.

**Vision Transformer for vision-based Deep RL**  Recent works, also compare the Vision Transformer to convolution-based architectures with a similar number of parameters and show that ViT is very data inefficient even when paired with an auxiliary task (Tao et al., 2022).

**Self-Supervised learning for image sequences**  Multiple works propose simple pretext tasks to train encoders to capture information from image sequences. These pretexts tasks can be playback speed classification (Yao et al., 2020), a temporal order classification (Misra et al., 2016; Lee et al., 2017; Xu et al., 2019), a jigsaw game (Ahsan et al., 2019) or a masked modelling task (Sun et al., 2019). A different approach consists of using contrastive learning. In this category, we can find methods that maximise the similarity between image sequences (Feichtenhofer et al., 2021), use autoregressive models to predict frames multiple steps in the future Lorre et al. (2020), and maximize the similarity between temporally adjacent frames (Knights et al., 2021).

## 3 BACKGROUND

### 3.1 VISION TRANSFORMER

ViT (Dosovitskiy et al., 2020) is a model, for image classification tasks, that doesn't rely on CNNs using only attention. The model wraps the encoder of a Transformer, uses patches of the input image as tokens and adds a classification token which after the computation will serve as the image representation. When compared to CNNs, ViT presents weaker image-specific inductive biases which can impact the sample-efficiency of the model during learning (d'Ascoli et al., 2021), however, it has been shown that with enough data the image-specific inductive biases become less important (Dosovitskiy et al., 2020). Moreover, ViT can capture relations between patches that are far apart from each other, thus deriving global information that can help the model perform better in certain tasks

### 3.2 REINFORCEMENT LEARNING

The problem of an **agent** learning to solve a task in a certain **environment** can be defined as a Markov Decision Process (MDP). A MDP $\mathcal{M}$ is defined by the tuple $\langle \mathcal{S}, \mathcal{A}, \mathcal{R}, \mathcal{T} \rangle$, where $\mathcal{S}$ is the set of states, $\mathcal{A}$ the set of actions, $\mathcal{R}$ the reward function, and $\mathcal{T}$ the transition function. At each timestep the agent is in a state $s \in \mathcal{S}$ and takes an action $a \in \mathcal{A}$. Upon performing the action the agent receives from the environment a reward $r \in \mathcal{R}$ and a new state $s' \in \mathcal{S}$ which is determined by the transition function $\mathcal{T}(s', s, a)$. The MDP assumes that the Markov property holds in the environment, i.e. the state transitions are independent and the agent only needs to know the current state to perform an action $P(a_t|x_0, x_1...x_t) = P(a_t|x_t)$. For the agent to decide what action to take it uses a policy function $\pi$, which gives a distribution over actions given a state, $\pi(a_t|s_t)$. This policy is evaluated using the function $V^\pi(s)$, which estimates the expected total discounted reward of an agent in a state $s$ and which follows a policy $\pi$.

### 3.2.1 DQN AND RAINBOW

DQN (Mnih et al., 2013) is a value-based method and uses a network with parameters $\phi$ that given a state $s$ outputs a prediction of the distribution of Q values over actions, $Q_\phi(s, a)$. The network learns the Q function by minimizing the mean squared error: $(y - Q_\phi(s, a))^2$, where $y = r + \gamma \, max_{a'} Q_\phi(s', a')$, as shown in Algorithm 1 at the Appendix.

Several works followed the DQN algorithm which introduced changes to improve performance. Rainbow (Hessel et al., 2017) combines six improvements, Double Q-Learning (van Hasselt et al., 2016), Prioritized Replay (Schaul et al., 2016), Dueling Networks (Wang et al., 2016), Multi-step Learning (Sutton & Barto, 2018), Distributional RL (Bellemare et al., 2017), and Noisy Nets (Fortunato et al., 2018) resulting in a more stable and sample efficient algorithm.

### 3.3 SELF-SUPERVISED METHODS

Recent self-supervised methods for vision tasks can be put in two main categories: contrastive and non-contrastive.

In contrastive learning, methods like MoCo (He et al., 2020) or SimCLR (Chen et al., 2020a) learn using a loss function that pulls the positive samples together and pushes the negative samples apart. These methods usually require very large batch sizes or auxiliary structures that allow for more negative samples. MoCo, in particular, has three iterations v1 (He et al., 2020), v2 (Chen et al., 2020b), and v3(Chen et al., 2021). In this work, we consider the more recent version (v3). This version uses a siamese network, where in one path the augmented samples (queries) are computed by an encoder $f_\theta$ (backbone) and a projector $g_\phi$, and in the other the samples (keys) by a momentum-encoder $f_\theta'$ and a projector $g_\phi$. The loss function is the InfoNCE loss, with temperature, of the dot product of the queries with the keys.

On the other hand, non-contrastive methods don't rely on the notion of positive and negative samples which results in a vast number of different approaches. DINO (Caron et al., 2021) consists of a siamese network where each path is fed with a random augmentation of the input and where the encoders learn to minimize the cross-entropy between their normalized output probability distributions, computed using a softmax with temperature scaling. The teacher encoder is updated using an exponential moving average of the student encoder parameters and in its computation path is used an additional centring operation that contributes to an asymmetry that helps the method avoid collapse. Unlike, most methods, DINO creates more than 2 augmentations of the same source. More precisely it creates a set of views composed of two global views and several local views. All views are computed by the student network while only the global views are computed by the teacher network, which pushes the student to create a local-to-global correspondence.

VICReg, on the other hand, tries to learn representations invariant to augmentations by minimizing the L2 distance while maintaining some variance in the representation features and decorrelating features. A more detailed explanation of the method will be presented in Section 4.

For this study we selected DINO, MoCo, and VICReg since they are currently considered state-of-the-art, their official implementations are available in PyTorch, and each represents a different type of approach.

## 4 TOV-VICREG

VICReg is a non-contrastive method that trains a network to be invariant to augmentations applied to the inputs while avoiding a trivial solution with the help of two additional losses, called variance and covariance, that act as regularizers over the embeddings. While VICReg is agnostic concerning the architectures used and even the weight sharing, in this work we consider the version where paths are symmetric, the weights are shared, and each path is composed of an encoder (also called backbone) and an expander. The expander increases the dimension of the representation vector in a non-linear way allowing the covariance loss to reduce dependencies and not only correlations of the representation vector. In addition, the expander also removes information that is not common to both representations.

VICReg uses three loss functions: **invariance** is the mean of square distance between each pair of embeddings from the same original image, as shown in Equation 1, where $Z$, and $Z'$ are two sets of embeddings, of size $N$, that result from computing two different augmentations of $N$ sources, and $z_j$ denotes the *j-th* embedding in the set; **variance** is a hinge loss that computes, over the batch, the standard deviation of the variables in the embedding vector and pushes that value to be above a certain threshold, as shown in Equation 2, where $d$ denotes the number of dimensions of the embedding vector, and $Z^j$ is the set of the *j-th* variables in the set of embedding $Z$; **covariance** is a function

that computes the sum of the squared off-diagonal coefficients of a covariance matrix computed over a batch of embeddings, as shown in Equation 3, to decorrelate the variables from the embedding. While the invariance loss function tries to make the model invariant to augmentations, i.e. output the same representation vector, the other two functions regularize the method by pushing the variables of the embedding vector to vary above a certain threshold and decorrelating the variables in each embedding vector.

$$i(Z, Z') = \frac{1}{N} \sum_{j}^{N} \left\| z_j - z_j' \right\|_2^2 \tag{1}$$

$$v(Z) = \frac{1}{d} \sum_{j}^{d} \max(0, \gamma - \sqrt{Var(Z^j)}) \tag{2}$$

$$c(Z) = \frac{1}{d} \sum_{i \neq j} [\text{Cov}(\text{Z})]_{i,j}^2 \tag{3}$$

TOV-VICReg or Temporal-Order-Verification-VICReg extends VICReg to better capture the temporal relations between consecutive observations and consequently encode extra information that can be useful in the deep reinforcement learning phase. To achieve that we add a new temporal order verification task, as seen in Shuffle-and-Learn (Misra et al., 2016), that consists of a binary classification task where a linear layer learns to predict if three given representation vectors are in the correct order or not. Like the other losses, we also employ a coefficient for the temporal loss and in most of our experiments, the value is 0.1. Figure 1 visually illustrates TOV-VICReg.

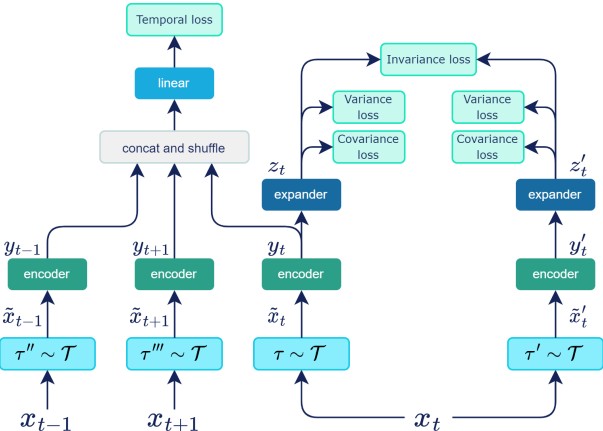

Figure 1: TOV-VICReg architecture

At each step we sample 3 consecutive observations, $\{x_{t-1}, \ x_t, \ x_{t+1}\}$, $x_t$ is processed by two different augmentations, and like VICReg these are the augmentations used in BYOL (Grill et al., 2020), while $x_{t-1}$ and $x_{t+1}$ are processed by two simple augmentations composed of a color jitter and a random grayscale. The $x_t$ augmentations are computed by the VICReg computation path and the resultant embeddings are used for the loss functions, i.e. variance, invariance, and covariance. In the **temporal order verification task** we encode the augmentation of $x_{t-1}$ and $x_{t+1}$, and concatenate those two representations with one of the representations of $x_t$, in our case we used the one that was augmented without solarize, obtaining the vector $\{y_{t-1}, y_t, y_{t+1}\}$. At last, we randomly permute the order of the representations in the vector and feed the resultant concatenated vector to a linear layer with a single output node that predicts if the given concatenated vector has the representations in the right order or not. The **temporal loss** used to optimize the model for this task is a Binary Cross Entropy loss. TOV-VICReg's pseudocode can be found in Appendix D.

## 5 PRE-TRAINING METHODOLOGY

We pretrained four encoders, one using our proposed method TOV-VICReg and three using state-of-the-art self-supervised methods: MoCov3 (Chen et al., 2021), DINO (Caron et al., 2021) and VICReg (Bardes et al., 2022). For this study, the encoder used is a Vision Transformer, more precisely the ViT tiny with a patch size of 8. We chose this patch size based on experiments that show that this value performed well in terms of data-efficiency when compared to 6, 10, and 12 without being too computationally intensive (Appendix B). Moreover, the implementation we use is an adaptation of the timm library (Wightman, 2019) implementation, which can be found in the source code of the DINO method. The dataset used is a set of observations from 10 of the 26 games in the Atari 100k benchmark, all available in the DQN Replay Dataset (Agarwal et al., 2020). For each game, we use three checkpoints with a size of one hundred thousand data points (observations), which makes up a total of three million data points (~55 hours). The pretraining phase is 10 epochs with two warmup epochs. We used the official code bases of all the self-supervised methods and tried to change the least amount of hyperparameters. Appendix H contains the tables with the hyperparameters used for each method.

## 6 DATA-EFFICIENCY

To test the pretrained Vision Transformers in reinforcement learning and compare data-efficiency gains, we trained in the 10 games used for pre-training for 100k steps using the Rainbow algorithm (Hessel et al., 2017), with the DER (van Hasselt et al., 2019) hyperparameters. The only difference between the agents at the start is the representation module. We chose two networks to compare against, the Nature CNN (Mnih et al., 2015), and a ResNet with an amount of parameters similar to ViT tiny (Appendix C) (Schwarzer et al., 2021b) that has a size roughly similar to the ViT tiny. Moreover, we use a learning rate two orders of magnitude smaller for the encoder ($1 \times 10^{-6}$), which previous works and experiments performed by us show to be beneficial (Schwarzer et al., 2021b).

In this section, to report our results we follow the rliable (Agarwal et al., 2021b) evaluation framework, where the scores of all games are normalized and treated as one single task.

### 6.1 RESULTS

Figure 2 shows the aggregate metrics of seven different encoders on 10 Atari games with training runs of 100k steps. The first four (ViT+<method>) are ViT tiny models pretrained with four different self-supervised methods, while the last three (ViT, ResNet, and Nature CNN) are randomly initialized models. Starting with the randomly initialized models we can assess that the Nature CNN and the ResNet are the most sample efficient models, with ViT far behind. Regarding the pretrained models, ViT, when pretrained with our method, performs better than the other models and the non-pretrained ViT in all metrics. It is worth noting that we report a higher variance in the results of our proposed method when compared to the remaining methods and non-pretrained models. ViT+TOV-VICReg when compared to Nature CNN, which has far fewer parameters, and ResNet, with a similar number of parameters seems to closely match their sample-efficiency performance (Appendix Table 7). Furthermore, the difference between the non-pretrained ViT and ViT pretrained with TOV-VICReg shows that a good self-supervised method that explores temporal relations and 3 million data points can help close the sample-efficiency gap while remaining a more complex and capable model. Regarding the remaining self-supervised methods, MoCo seems to perform considerably well obtaining even a median very similar to TOV-VICReg and is then followed by DINO and VICReg, respectively. All pretrained ViTs show an improvement in comparison to the non-pretrained ViT.

## 7 METRICS

A significant phenomenon when doing self-supervised training is the collapse of the representations, which can be seen in three forms: representational collapse, dimensional collapse, and informational collapse. Representational collapse refers to the features of the representation vector collapsing to a single value for every input, meaning the variance of the features is zero, or close to zero. In

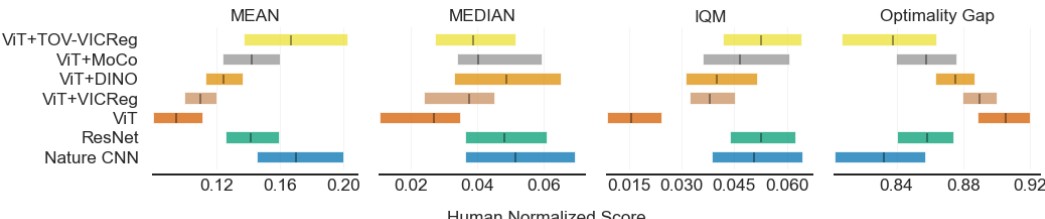

Figure 2: The eval runs across the different games are normalized and treated as a single task. The IQM corresponds to the Inter-Quartile Mean among all the runs, where the top and bottom 25% are discarded and the mean is calculated over the remaining 50%. The Optimality Gap refers to the number of runs that fail to surpass the human average score, i.e. 1.0.

dimensional collapse, the representations don't use the full representation space, which can be measured by calculating the singular values of the covariance matrix calculated over the representations. Informational collapse defines the case where the features of the representation vector are correlated and therefore are representing the same information.

**Dimensional Collapse** All methods seem to avoid dimensional collapse, i.e. most dimensions have a singular value larger than zero, as observed in Figure 3. However, we notice that some methods make better use of the space available since they present higher singular values. TOV-VICReg, in particular, seems to excel in this metric, even improving the results obtained by VICReg. It is worth noting that both VICReg and TOV-VICReg employ a covariance loss that helps decorrelate the embedding variables which may be contributing positively to these results. Furthermore, we used a covariance coefficient of 10 for TOV-VICReg and 1 for VICReg a change that according to our experiments culminates in the increase here observed.

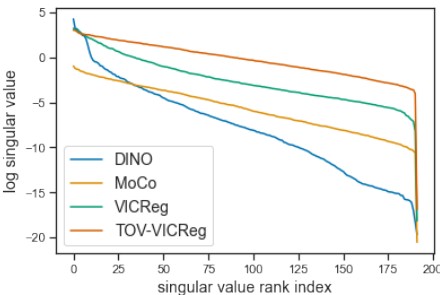

Figure 3: Logarithm of the singular values of the representation vector's covariance matrix sorted by value.

**Representational Collapse** Results in the first row of Table 1 show the computed standard deviation of the representation vector over a batch of thousands of data points. DINO, VICReg and TOV-VICReg show a value well above zero, meaning that none of the methods suffered from representation collapse during training. On the other hand, MoCo shows a much smaller value of 0.178, which is still, is far from a complete collapse. Both VICReg and TOV-VICReg use a hinge loss that pushes the representation vector to have a standard deviation of 1 or above. While VICReg slowly converges to this value our method converges to roughly 1.65, which might be the result of adding a temporal order verification task.

**Informational Collapse** We report in the second row of Table 1, the comparison of the average correlation coefficients of the representation vectors. TOV-VICReg performs better than the other methods, including VICReg, which present very similar coefficients. Like in the dimensional collapse, this result is in part due to the higher covariance coefficient used in TOV-VICReg which by design

| Metric | DINO | MoCo | VICReg | TOV-VICReg |
|---|---|---|---|---|
| Std | 0.979 | 0.178 | 1.003 | 1.648 |
| Corr. Coef. | 0.1764 | 0.1538 | 0.1531 | 0.0780 |

Table 1: Average standard deviation and correlation coefficient of the representation vector

helps the model to decorrelate the representation's features. Increasing the coefficient in VICReg results in a lower correlation coefficient as well, but is still higher than TOV-VICReg.

## 8 REPRESENTATIONS

In this section, we present different visualizations to better understand the representations learned by each of the methods. Our goal with the following visualizations is to help us better understand the learned representations and give some intuitions about their properties.

**Cosine similarity** Figure 4 presents a similarity matrix of the representations where we can observe that TOV-VICReg can better distinguish between observations of different games but also observations from the same game, as shown in Figure 5. MoCo, on the other hand, seems to make a good distinction between observations from the same game. However, we can observe in the colour bar that all the representations are very similar to each other, which corroborates the results obtained in Section 7. Oppositely, VICReg and DINO manage to spread representations more, as we can see in the colour bars, but, the yellow squares in the diagonal show that the representations from the same game are more similar to each other which is corroborated by Figure 5. Given the empirical results, we believe that this capacity to distinguish observations from the same game might be a good indicator.

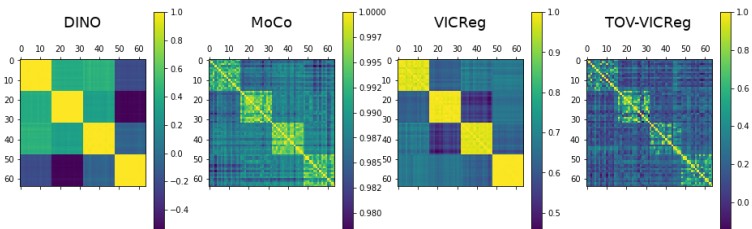

Figure 4: Similarity matrices of the representations computed by MoCo, DINO, VICReg, and TOV-VICReg respectively. There are a total of 64 data points, from 4 different games: Alien, Breakout, MsPacman, and Pong, where from 0-15 are from Alien, 16-31 are from Breakout and so forth.

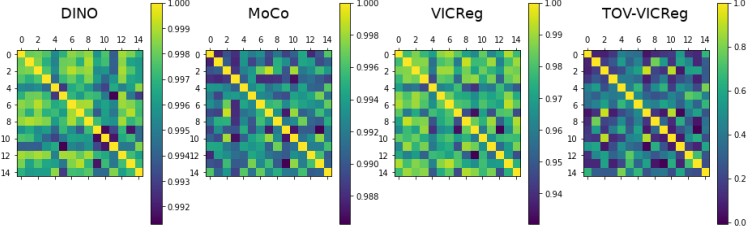

Figure 5: Similarity matrices of the representations computed by MoCo, DINO, VICReg, and TOV-VICReg respectively, of observations from MsPacman.

**Attention visualisation** The research work that proposes DINO shows that the Vision Transformer is able to attend to important parts of the input after training using DINO. Inspired by these results, we try to make the same evaluation for the several self-supervised methods we are studying, including TOV-VICReg, and try to understand if any of the encoders can attend to interesting parts of the input.

In Figure 6, we can see the results of all methods for an observation from the game of Pong, where each method produces three attention maps, one for each self-attention head of the last block of the Vision Transformer. All pretrained ViT seem to attend at some level to important game features like the ball and the paddles. However, TOV-VICReg is the only method that doesn't spread the attention to other parts of the frame that we don't consider important to describe the current state of the game. When comparing to VICReg's attention maps we believe that the temporal order verification task greatly helped the attention of the model. In more visually complex games, e.g. Freeway or MsPacman, these attention maps start to be more difficult to analyse but it is still possible to discern some important features.

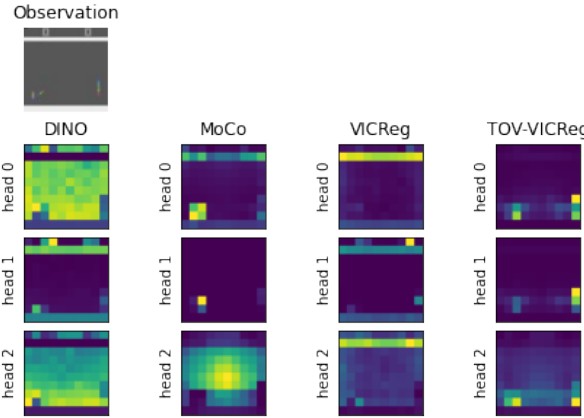

Figure 6: Attention maps produced by the pretrained ViTs. We fed a pretrained ViT with an observation from the game Pong and obtained the attention maps from the three heads in the last block.

## 9 DISCUSSION & CONCLUSION

In this work, we presented a study of ViT for vision-based deep reinforcement learning using self-supervised pretraining, and proposed a self-supervised method that extends VICReg to better capture temporal relations between consecutive observations. Our results showed that the agent using a Vision Transformer that was pretrained with our method manages to surpass all other Vision Transformers, pretrained and non-pretrained, in sample efficiency and also achieves results very close to convolution-based models with far fewer parameters. These results reinforce the importance of encoding temporal relations between observations in the representation model, as shown by previous works, and also show that even vision models with weaker inductive biases and more parameters, when well pretrained, can achieve similar results in sample efficiency. Furthermore, we show several metrics, evaluation tasks and visualizations which can be of great value for future work.

The ability to use larger models, with millions of parameters, that are as sample efficient as some of the most popular CNN-based models (like Nature CNN or Impala ResNet), with thousands of parameters, is very important since it opens the door to using Deep RL in even more complex problems where smaller models tend to struggle to perform, without losing sample-efficiency. In this work, we try to advance the knowledge by studying the pretrain of a vision transformer using self-supervised methods. This approach has seen successes in natural language processing (Devlin et al., 2019; Brown et al., 2020), and computer vision (Radford et al., 2021) and we believe that similar approaches in RL have the potential to unlock new levels of performance never achieved before (Baker et al., 2022).

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
