# OpenReview forum: "Pretraining the Vision Transformer using self-supervised methods for vision based Deep Reinforcement Learning"
_ICLR.cc/2023/Conference — Submitted to ICLR 2023_

### Official Review · Reviewer_X82N · 2022-10-23

**Confidence:** 4
**Correctness:** 3
**Technical Novelty And Significance:** 2
**Empirical Novelty And Significance:** 2
**Recommendation:** 3

**Clarity, Quality, Novelty And Reproducibility:**

### Quality

This paper should be improved in terms of clairty and experiments before being accepted into ICLR.

### Novelty:

The paper presents an incremental contribution where ViTs are trained by combining the temporal order verification loss (Misra et al, 2016) and the VICReg loss (Bardes et al, 2021). Both the methods and the results do not present significantly contributions.

### Reproducibility

The authors provided their code in the supplemental material.

### Clarity

1. Figure 4 and 5 are confusing becuase of the different scales of the color map.
2. The desription of the baseline methods is not clear. For the SGI ResNet Large, was it pretrained with self-supervision loss? If so, did the pretraining follow the original SGI paper?

Minor Comments:

1. The DQN algorithm details on page 3 can be presented in an appendix and formatted with Latex algorithm packages.
2. VICReg is a published ICLR 2022 paper. Please update the reference accordingly.
3. For audience unfamiliar with VICReg, it would be helpful to explain how expanders work.

**Strength And Weaknesses:**

### Strength

1. The proposed self-supervised learning method is well suited for RL tasks and experiment results show that it improves performance of self-supervised learning methods that does not capture the temporal relationship.

### Weaknesses

1. This paper lacks novelty in terms of the proposed self-supervised learning method. Also, the presented results are mostly well-known in the community.
2. Temporal order verification is one type of self-supervision loss that aims to capture temporal dependency. This paper does not justify the choice of Shuffle-and-Learn against other methods [1, 2, 3]. I’d recommend the authors to add a discussion on relevant self-supervised learning literature for temporal relationship learning and provide a justification on the choice of Shuffle-and-Learn.
3. The number of steps used for temporal order verification loss is an important hyperparameter that would affect the quality of the learned representations. This paper does not provide experimental results showing its impact. I’d encourage the authors to add an experiment varying the number of steps.

### References

[1] Lee, Hsin-Ying, et al. "Unsupervised representation learning by sorting sequences." *Proceedings of the IEEE international conference on computer vision*. 2017.

[2] Xu, Dejing, et al. "Self-supervised spatiotemporal learning via video clip order prediction." *Proceedings of the IEEE/CVF Conference on Computer Vision and Pattern Recognition*. 2019.

[3] Yao, Yuan, et al. "Video playback rate perception for self-supervised spatio-temporal representation learning." *Proceedings of the IEEE/CVF conference on computer vision and pattern recognition*. 2020.

**Summary Of The Paper:**

This paper studies if Vision Transformers (ViTs) could outperform CNNs in vision-based RL tasks. The authors compared existing self-supervised strategies and proposed a new self-supervised approach designed for the sequential observations in RL. The newly proposed method combines previous ideas from VICReg (Bardes et al, 2021) and Shuffle-and-Learn (Misra et al, 2016). The authors show that the new approach TOV-VICReg outperforms other SOTA self-supervised learning methods in the RL setting. However, from the authors experiments, CNNs still outperform ViTs trained with TOV-VICReg in eight out of the ten tested Atari games.

**Summary Of The Review:**

I’m inclined to reject this paper mainly because: 1) the experiments do not justify the motivation of this work, i.e., use large models to solve complex RL tasks where smaller models struggle, as stated in the conclusion of this paper. Clearly, the chosen environments are not complex enough to demonstrate the potential benefits of pretrained ViTs. 2), the proposed method lack novelty and the results do not bring significant new insights to the community, 3) the paper does properly compare other self-supervised learning methods that aim to capture the temporal relations  4) the paper presentation needs further polishing.

---

> ### Author Response · Authors · 2022-11-15
> **Submision update and comments after feedback**
>
> First of all, thank you so much for your review and the time you spend on it. We have updated our paper and appendix to incorporate the feedback we received. All the new changes are highlighted in blue.
>
> **"Also, it is well-known that the use of pre-training is helpful for vision-based reinforcement learning."** - The success of pre-trained encoders for vision-based reinforcement learning based on self-supervised learning methods that only use observations is in our opinion not clear:
> - Schwarzer et al., 2021b (SGI) pretrain a ResNet with Self-Predictive Representations and two auxiliary objectives: Goal-Conditioned Reinforcement Learning, and Inverse Dynamics Modeling, which use rewards and actions, respectively, from the dataset. Also, the authors show that their method without Inverse Dynamics Modeling suffers from representational collapse.
> - Zhan et al., 2020 (FERM) pretrain a convolutional encoder using a contrastive loss, however, this work was proposed for a real-robot arm setting, which is significantly different visually when compared to Atari games observations.
> - Stooke et al. (2021) pretrain a convolutional encoder using a Spatio-temporal contrastive loss but show only marginal improvements in Atari games.
> - Anand et al. (2019) pretrain a convolutional encoder using spatio-temporal contrastive loss called SpatioTemporal DeepInfoMax in observations from Atari games. However, it’s unknown if the pretrained encoder resulted in better performance or data-efficiency since they never tested the pretrained encoder in reinforcement learning. Instead, they tested the encoder in a new linear probing task called AtariARI, however, this can only be used for a limited number of Atari games.
>
> In our work we pretrain the Vision Transformer using several self-supervised methods. We show that all methods avoid representational collapse and that one of the simplest pretext tasks for image sequences is able to largely outperform the non-pretrained ViT and match a much smaller and more sample efficient model (Nature CNN). Finally, we re-added (removed before submission)  a simple linear probing task based on imitation learning (i.e. predicting an action given an observation) that can be used in any environment and not just a few Atari games. This task was very helpful in evaluating the quality of the pretrained encoders during the development without having to resort to reinforcement learning and for that reason, we believe it can be useful for future work (code available in the supplements).
>
> "Temporal order verification is one type of self-supervision loss that aims to capture temporal dependency. This paper does not justify the choice of Shuffle-and-Learn against other methods [1, 2, 3]. I’d recommend the authors to add a discussion on relevant self-supervised learning literature for temporal relationship learning and provide a justification on the choice of Shuffle-and-Learn." - We update our related work to address your comment.
>
> **"The number of steps used for temporal order verification loss is an important hyperparameter that would affect the quality of the learned representations. This paper does not provide experimental results showing its impact. I’d encourage the authors to add an experiment varying the number of steps."** - We have further experiments in procgen games that show the limitations of using consecutive frames for environments where backgrounds are dynamic. We tested changing the distance of the consecutive frame to x_{t+k} with k=6 and k=12 and observed no improvement in our evaluation task. However, it is possible that this result in improvements when used in Reinforcement Learning but we are unable to report those results in time for this rebuttal.

---

### Official Review · Reviewer_Wxq5 · 2022-10-29

**Confidence:** 3
**Correctness:** 2
**Technical Novelty And Significance:** 3
**Empirical Novelty And Significance:** 2
**Recommendation:** 3

**Clarity, Quality, Novelty And Reproducibility:**

Clarity: I am not very sure about the RL protocol. Is the representations trained in an offline setting that pre-stores frames?
Quality: not good enough for both technical depth or presentation quality.
Novelty: The training framework is somewhat new.
Reproducibility: Codes are provided.

**Strength And Weaknesses:**

Strengths:
+ The proposed training framework that combines VICReg and temporal verification is new to my knowledge.
+ Experimental results show that the variant out-performs alternative representation learning methods in the inspected RL setting.
+ A set of analysis shows that the method shows better statistical properties w.r.t. collapse and feature similarity.

Weaknesses:
- Most of the experimental results seem not related to reinforcement learning and the only RL evaluation reports inferior performance than (randomly initialized?) CNNs. I recommend the authors to consider generic video classification evaluation protocols in [A], which may be a more promising thing to pursue.
- The language is fine but the organization is pool IMHO. Too much space is spent on describing standard things like MDP, DQN or VICReg.
- Generally I am not convinced by the motivation. I think transformers are widely considered less data-efficient and I don't understand why authors expect it to behave the opposite way in reinforcement learning.
- Typos:
components that allows us -> allow
the increasing interest -> and the increasing interest
ViT presents weaker image-specific inductive biases which allow the CNNs for much sample-efficient learning; allows CNNs?? what does this mean

[A] VideoMoCo: Contrastive video representation learning with temporally adversarial examples, CVPR 2021

**Summary Of The Paper:**

*I am a vision person and an emergency reviewer.*

This paper proposes to combine VICReg and a temporal verification loss for video representation learning, and inspect its impact on the data-efficiency of down-streaming reinforcement learning tasks. It is shown that the proposed method out-performs VICReg, MOCO and DINO while still under-performing (randomly initialized?) CNNs. Further analyses show that the proposed variant of VICReg shows better properties in terms of collapse and feature distribution.

**Summary Of The Review:**

This is an empirical study yet I am not convinced by the motivation or the significance of empirical results.

---

> ### Author Response · Authors · 2022-11-15
> **Submision update and comments after feedback**
>
> First of all, thank you so much for your review and the time you spend on it. We have updated our paper and appendix to incorporate the feedback we received. All the new changes are highlighted in blue.
>
> **"Most of the experimental results seem not related to reinforcement learning and the only RL evaluation reports inferior performance than (randomly initialized?) CNNs"** - We have updated the section to improve its clarity regarding the models.
>
> **" I recommend the authors to consider generic video classification evaluation protocols in [A], which may be a more promising thing to pursue."** - Thank you, for your recommendation, given our results we agree with you and find that exploring such methods should an interesting topic for future work. However, we won't be able to include such experiments since it is meet the goal of this work.
>
> **"The language is fine but the organization is pool IMHO. Too much space is spent on describing standard things like MDP, DQN or VICReg."** - We try to make a good introduction to the most important pieces of our work so that readers can better understand our work.
>
> **"Typos: components that allows us -> allow the increasing interest -> and the increasing interest ViT presents weaker image-specific inductive biases which allow the CNNs for much sample-efficient learning; allows CNNs?? what does this mean"** - We have addressed your comments in the updated version.
>
> **"Generally, I am not convinced by the motivation. I think transformers are widely considered less data-efficient and I don't understand why authors expect it to behave the opposite way in reinforcement learning."** - We don’t expect them to be less data-efficient we assume that they are less data-efficient, our goal is to understand good pretraining approaches that are able to produce more sample efficient

---

### Official Review · Reviewer_Wuo3 · 2022-10-30

**Confidence:** 3
**Correctness:** 3
**Technical Novelty And Significance:** 2
**Empirical Novelty And Significance:** 2
**Recommendation:** 3

**Clarity, Quality, Novelty And Reproducibility:**

The clarity and writing quality of the paper can be improved. One example, the mode performance analysis in Sec 6.1 is unclear. It is not self-contained enough for one to understand the metric. And it is confusing if the results focus only on “sampling efficiency” or actually looks at the model’s game performance.

The papers have a few broken links.

While it could claim the method is novel for RL, but the specific connection between the proposed method and RL is not clear to me. And a generic method to pretrain a feature extractor on video. Such a combination of image-only objectives and temporal ordering prediction is marginally novel.

The amount of details in the paper is sufficient for one to understand the main idea but may not be sufficient for one to reproduce the results fully.

**Strength And Weaknesses:**

** Strength: **
1. It is reasonable to combine the image only pretraining method (specifically, VICReg) with temporal ordering prediction as the overall objectives covers both image and temporal information.
2. The paper analyzes the property of representation from a few different aspects in the experiments.

** Weaknesses: **
1. It is unclear how the pretraining is specific to deep reinforcement learning. It looks like a generic video-based pretraining method, and its connection with reinforcement learning seems a bit loose.
2. The novelty of the method is limited as combining two established self-supervised learning methods in the same training objective is not something difficult to come up with.
3. Q: it seems Atari 100K has 26 games (please correct me if I am wrong), why only 10 games are used?
4. The experimental results are not strong. Especially, using visual transformer does not seem to be that competitive to using CNN, as the paper claimed.
5. The clarity of the paper (especially in the experiment section) can be improved.

**Summary Of The Paper:**

This paper aims to introduce visual transformers to extract feature representations for reinforcement learning. It proposed a method, which is a combination of VICReg and temporal ordering prediction, to pretrain the ViT network. Experiments are done on Atari 100K using the visual transformer pretrained by the proposed method.

**Summary Of The Review:**

The paper has a good motivation of using a well-pretrained visual transformer in deep RL. However, the method design seems to be generic and of marginal novelty. The experimental results are a bit confusing and do not seem to be staring enough to address the paper’s motivation.

---

> ### Author Response · Authors · 2022-11-15
> **Submision update and comments after feedback**
>
> First of all, thank you so much for your review and the time you spend on it. We have updated our paper and appendix to incorporate the feedback we received. All the new changes are highlighted in blue.
>
> **"It is unclear how the pretraining is specific to deep reinforcement learning. It looks like a generic video-based pretraining method, and its connection with reinforcement learning seems a bit loose."** - We agree with your assessment that TOV-VICReg is not specific to reinforcement learning. While we could have used methods that use rewards and actions during the pre-training phase, for example, SGI, we decided to only consider methods that only use images. This allows using pre-training in environments that don’t have a good Offline RL dataset as we have for Atari, which we consider to be rather important for possible future applications.
>
> **"The novelty of the method is limited as combining two established self-supervised learning methods in the same training objective is not something difficult to come up with."** - We agree with you that our method is not very novel however that was not our goal for this work. Our goal was to study pretraining a Vision Transformer using several SOTA self-supervised learning methods for images in a vision-based reinforcement learning setting. Observations from Atari games are very different from the natural images we can find in datasets like ImageNet and it is not very well understood how these SOTA methods perform when used in a dataset where the images are much simpler and smaller. Our work shows a comparison of the SOTA methods and our simple method (TOV-VICReg) using a set of tools: data-efficiency in reinforcement learning, metrics to evaluate representation collapse, and visualizations that help us understand why some encoders are better than others. Additionally, we show great advantages in pretraining ViT, especially with our method which considers temporal relations between observations. Furthermore, we re-added (removed before submission) a simple linear probing task based on imitation learning (i.e. predicting an action given an observation) that can be used in any environment and not just a few Atari games, like AtariARI (Anand et al. (2019)). This task was very helpful in evaluating the quality of the pretrained encoders during the development without having to resort to reinforcement learning and for that reason, we believe it can be useful for future work (code available in the supplements).
>
> **"it seems Atari 100K has 26 games (please correct me if I am wrong), why only 10 games are used?"** - You are correct, given the number of different approaches here tested, using 26 games would be unfeasible given our computation resources. We tested pretraining TOV-VICReg in the Atari100k 26 games and evaluated them in the same 10 games and didn’t observe any improvements. Furthermore, since we weren’t aiming for SOTA performance we decided only to use 10 games which we considered to be enough.
>
> **"The clarity of the paper (especially in the experiment section) can be improved."** - We have updated the paper to improve this section.
>
> "The papers have a few broken links." - Can you be more specific about which links are broken? Also, we have several links for the Appendix which is available in the supplementary materials.

---

> > ### Comment · Reviewer_Wuo3 · 2022-12-03
> > **Thanks**
> >
> > Thank you for the authors' response. Overall, I would say the content of the response is understandable but not convincing to change my evaluation of the paper.

---

### Official Review · Reviewer_9PJR · 2022-10-31

**Confidence:** 3
**Correctness:** 4
**Technical Novelty And Significance:** 2
**Empirical Novelty And Significance:** 4
**Recommendation:** 3

**Clarity, Quality, Novelty And Reproducibility:**

This paper is generally clear. The novelty is limited due to no originated method is proposed in this paper. The TOV-VICReg is a combination of two existing methods. The proposed method should be easy to reproduce based on the authors' code.

**Strength And Weaknesses:**

Strength:
1. The authors present interesting and trials with ViTs for RL.

Weaknesses:
1. The novelty is limited. The proposed method is made up of several previous methods.
2. Even though the proposed TOV-VICReg is more effective than the other self-supervised methods, it is still only comparable to the CNN based models with no significant advantage. In other words, the ViT based models remain impractical for RL.
3. It would be better if the authors can consider reconstruction based self-supervision like MAE and SimMIM in their experiments, which adopts a quite different objective compared with DINO and MoCo.


**Summary Of The Paper:**

This paper mainly targets usage of ViTs in Reinforcement Learning. The authors conduct several experiments with different training strategies for ViTs, showing the effectiveness of self-supervised learning for ViTs in RL. Besides, the authors propose a new self-supervised objective function based on VICReg, which involves the temporal order prediction for learning better temporal relations.

**Summary Of The Review:**

The authors present good technical analysis of using ViTs for RL problems. The novelty and significance are limited.

---

> ### Author Response · Authors · 2022-11-15
> **Submision update and comments after feedback**
>
> First of all, thank you so much for your review and the time you spend on it. We have updated our paper and appendix to incorporate the feedback we received. All the new changes are highlighted in blue.
>
> **"The novelty is limited. The proposed method is made up of several previous methods."** - We agree with your assessment that our method is not very novel however that was not our goal for this work. Our goal was to study pretraining a Vision Transformer using several SOTA self-supervised learning methods for images in a vision-based reinforcement learning setting. Observations from Atari games are very different from the natural images we can find in datasets like ImageNet and it is not very well understood how these SOTA methods perform when used in a dataset where the images are much simpler and smaller. Our work shows a comparison of the SOTA methods and our simple method (TOV-VICReg) using a set of tools: data-efficiency in reinforcement learning, metrics to evaluate representation collapse, and visualizations that help us understand why some encoders are better than others. Additionally, we show great advantages in pretraining ViT, especially with our method which considers temporal relations between observations. Furthermore, we re-added (removed before submission) a simple linear probing task based on imitation learning (i.e. predicting an action given an observation) that can be used in any environment and not just a few Atari games, like AtariARI (Anand et al. (2019)). This task was very helpful in evaluating the quality of the pretrained encoders during the development without having to resort to reinforcement learning and for that reason, we believe it can be useful for future work (code available in the supplements).
>
> **"Even though the proposed TOV-VICReg is more effective than the other self-supervised methods, it is still only comparable to the CNN based models with no significant advantage. In other words, the ViT based models remain impractical for RL."** - While we agree with your assessment we would like to highlight three things:
> - Our method builds on top of VICReg which ended up being the worst-performing method, it’s likely that using a similar pretext task with MoCo, for example, can obtain even better results.
> - Other pretext tasks for image sequences have been proposed, for example, PacePred (Wang et al. 2020, Self-supervised Video Representation Learning by Pace Prediction), which show better performance than Shuffle&Learn. We believe that future work can explore using some of those methods to improve our results.
> - In this work, we only explore Atari games, which are far from being visually complex. Studying a similar approach in environments that give visually complex observations, for example, natural images like in FERM (Zhan et al., 2020, Learning Visual Robotic Control Efficiently with Contrastive Pre-training and Data Augmentation) can show advantages.
>
> **"It would be better if the authors can consider reconstruction based self-supervision like MAE and SimMIM in their experiments, which adopts a quite different objective compared with DINO and MoCo."** - Thank you for your feedback, we agree with your statement and we think it would be of great value to consider one method from this category. Unfortunately, we find that more than four methods would make the paper more difficult to understand and some results more difficult to report and for those reasons, we have to leave the study of such methods for future work.

---

### Official Review · Reviewer_rKFV · 2022-11-01

**Confidence:** 4
**Correctness:** 3
**Technical Novelty And Significance:** 2
**Empirical Novelty And Significance:** 2
**Recommendation:** 3

**Clarity, Quality, Novelty And Reproducibility:**

- The core contribution is not very clear to me.
- Regarding the use of self-supervised learning for RL, "CURL: Contrastive Unsupervised Representations for Reinforcement Learning" is one of the work that popularized this. The authors should cite and compare against this work.
- In light of the existing work, more work should be done to advance the field. Just trying self-supervised learning on reinforcement learning is not sufficient for publication at ICLR.
- The contribution may be the use of self-supervised learning on ViT based image RL. The proposed method does not contain modifications for ViTs: TOV and VICReg are very generic methods, and they can be applied to any deep nets that take images (regardless of CNN vs ViT, RL vs supervised learning problems). Can authors provide why the proposed method is significantly novel or overcomes challenges in using self-supervised learning for RL?
- Also, could authors provide the results of applying the proposed methods on CNNs? If the CNN performance can't be improved, why so?

**Strength And Weaknesses:**

[Strengths]
- The paper is easy to follow.
- The proposed method makes intuitive sense to me.

[Weaknesses]
- Most importantly, the main contribution of the submission is not very clear to me.
- Also, the technical novelty and empirical novelty is quite limited. TOV and VICReg are existing self-supervised learning methods. Also, it is well-known that the use of pre-training is helpful for vision-based reinforcement learning.
- The analysis in Section 7 (representation collapse, dimension collapse, etc) are very generic and do not provide much insight on using self-supervised learning for reinforcement learning.
- It seems that the CNN result is without self-supervised training and ViT result is with self-supervised training (although it is unclear to me). Can authors add more on this?
- I don't see the learning curve of return vs environment steps. This is pretty important for demonstrating sample efficiency in RL.

**Summary Of The Paper:**

The paper studies the effect of self-supervised pre-training for Vision Transformer based RL agents. It shows the effect of temporal order verification and VICReg on 10 Atari games. The proposed methods improves final return on Atari games.

**Summary Of The Review:**

The submission is limited in terms of both theoretical novelty and empirical novelty. Especially, the empirical validity is pretty limited due to small set of experiments, missing experiment details, etc. More work should be done to warrant the publication of this work.

---

> ### Author Response · Authors · 2022-11-15
> **Submision update and comments after feedback**
>
> First of all, thank you so much for your review and the time you spend on it.
> We have updated our paper and appendix to incorporate the feedback we received. All the new changes are highlighted in blue.
>
> **"the main contribution of the submission is not very clear to me."** - The main contribution of this paper is the study of pretraining a vision transformer using self-supervised learning methods for images and a simple extension of VICReg that captures temporal relations, in the context of reinforcement learning or more specifically Atari games.  Our results show relevant data efficiency gains, significant benefits in temporal relations, and that all SSL are able to avoid collapse. We also use a set of metrics and visualizations, that are of great value for future work in order to compare and measure the quality of the representations learned by future methods. To reinforce the necessity of better benchmarks for this type of approach we have re-added (removed before submission) a simple linear probing task based on imitation learning. We found it helpful in evaluating the quality of the pretrained encoders during the development without resorting to reinforcement learning. We believe it can be helpful for future work.
>
> **"- also the main contribution of the submission is not very clear to me."** - Our aim with this paper was not to develop a completely novel method but instead to study pretraining ViT with generic self-supervised learning methods for images in the context of vision-based deep reinforcement learning. Adding TOV to VICReg serves as a simple example of exploiting temporal relations which helps us demonstrate the importance of using these relations. Using more complex pretext tasks or even spatio-temporal methods (which we do not explore), will most likely improve the results here presented.
>
> **"The analysis in Section 7 (representation collapse, dimension collapse, etc) are very generic and do not provide much insight on using self-supervised learning for reinforcement learning."** -  As we stated in the paper it is common for the encoder to collapse during self-supervised pre-training. In section 7 we present several metrics that allow us to assess different types of collapses and to the best of our knowledge, this is the first paper to report such evaluation in observations from RL environments. While we present these values after the reinforcement learning results, it is of great value to compute them after or even during the pre-training.
>
> **"It seems that the CNN result is without self-supervised training and ViT result is with self-supervised training (although it is unclear to me). Can authors add more on this?"** - We have updated the paper to be clearer in this regard.
>
> **"I don't see the learning curve of return vs environment steps. This is pretty important for demonstrating sample efficiency in RL."** - Thank you for your feedback, while some papers use learning curves to report data-efficiency gains we think that these are not as valuable and contain too much noise. To report our results we used the methodology proposed by (Agarwal et al 2021, Deep Reinforcement Learning at the Edge of the Statistical Precipice) using the RLiable library.

---

### Decision · Program_Chairs · 2023-01-20

**Decision:**

Reject

**Justification For Why Not Higher Score:**

All reviewers feel negative on the current work, the ac feels the same.

**Justification For Why Not Lower Score:**

N/A

**Metareview: Summary, Strengths And Weaknesses:**

All reviewers feel negative about the current submission. The raised issues include limited main contribution and technical novelty, unclear presentation, and experimental validations. Although the authors try to address these issues, the raised novelty issue is hardly solved. After checking all the reviews and rebuttals, the AC feels the current work needs a significant revision to address the raised issues. The authors shall take these suggestions to make further improvements.